# The Antifungal Effect of Gaseous Ozone on *Lasiodiplodia theobromae* Causing Stem-End Rot in ‘Keitt’ Mangoes

**DOI:** 10.3390/foods12010195

**Published:** 2023-01-01

**Authors:** Nonjabulo L. Bambalele, Asanda Mditshwa, Nokwazi C. Mbili, Samson Z. Tesfay, Lembe S. Magwaza

**Affiliations:** 1Department of Horticultural Sciences, School of Agricultural, Earth and Environmental Sciences, University of KwaZulu-Natal, Pietermaritzburg 3209, South Africa; 2Department of Plant Pathology, School of Agricultural, Earth and Environmental Sciences, University of KwaZulu-Natal, Pietermaritzburg 3209, South Africa; 3Department of Crop Sciences, School of Agricultural, Earth and Environmental Sciences, University of KwaZulu-Natal, Pietermaritzburg 3209, South Africa

**Keywords:** ozone, stem-end rot, peroxidase activity, mango, flavonoids

## Abstract

This study evaluated the antifungal activity of ozone (O_3_) against stem-end rot of mango fruit (*cv*. Keitt). Mango fruit were exposed to gaseous ozone (0.25 mg/L) for 24 or 36 h during cold storage, and control fruit were untreated. Experimental fruit were stored at 90% relative humidity and 10 ± 0.5 °C for three weeks and ripened at ambient temperature for one week. Ozone treatment (24 h) inhibited the mycelial growth of *Lasiodiplodia theobromae* by 60.35%. At day twenty-eight of storage, fruit treated with O_3_ for 36 h had low mass loss (%) and high firmness compared to the untreated control fruit. Treating mango fruit with O_3_ (36 h) maintained the color and concentration of total flavonoids throughout the storage time. At the end of storage, peroxidase activity under the O_3_ 24 h treatment was significantly higher (0.91 U min^−1^ g^−1^ DM) compared to O_3_ (36 h) and control, which, respectively, had 0.80 U min^−1^ g^−1^ DM and 0.78 U min^−1^ g^−1^ DM. Gaseous ozone for 24 h is recommended as a cost-effective treatment for controlling stem-end rot. These findings suggest that gaseous ozone effectively controlled stem-end rot and enhanced the postharvest quality of mango fruit.

## 1. Introduction

*Lasiodiplodia theobromae* is a fungus that causes stem-end rot in mango fruit during postharvest storage, leading to significant economic losses [1]. The *L. theobromae* infection occurs preharvest; the pathogen pierces the stem through mechanical injuries, lenticels, and other natural openings [2]. The fruit remains asymptomatic until postharvest, when storage conditions change and ripening occurs [3]. The symptoms include dark brown and water-soaked spots at the fruit stem end, which become severe towards fruit senescence [2]. In South Africa, fungicides such as prochloraz and imazalil are generally used to control postharvest diseases, including stem-end rot [4]. However, there is apprehension about the production of toxins, including haloacetic acids and trihalomethanes, pathogen resistance against fungicides, and chemical residues in the fresh produce industry [5,6]. This has led to research focusing on eco-friendly postharvest technology such as hot water, ozone (O_3_), edible coatings, and UV-C treatments.

Ozone is generally recognized as safe, with powerful oxidation potential and antimicrobial properties [7]. It is produced through corona discharge or photochemical (UV) and can be applied as an aqueous solution or gas [8]. Corona discharge is mainly used as a postharvest treatment for fresh produce [9]. Ozone is then decomposed rapidly to form oxygen molecules, leaving no chemical residues on fresh produce [7]. Ozone is produced continuously during the storage of fresh produce and cannot be accumulated [9]. The advantage of O_3_ as a postharvest treatment is that it is effective against various microorganisms at minute concentrations with low energy input [6]. Thus, O_3_ is an environmentally friendly and economical postharvest treatment.

Gaseous ozone treatment at 2.14 µg m^−3^ reduced the incidence of *Fusarium proliferatum* in ‘Morado de Cuenca’ garlic bulbs during storage at 22 °C for fourteen days [10]. An earlier study by Tzortzakis et al. [11] reported that O_3_ (0.05 µmol mol^−1^) decreased the lesion diameter in the ‘Mareta’ tomato fruit inoculated with *Botrytis cinerea.* Ong and Ali [12] observed that ozone at a concentration of 2.5 µL/L inhibited spore germination, disease incidence, and severity of *Colletotrichum gloeosporioides* in ‘Sekaki’ papaya stored at ambient temperature for fourteen days [12]. Experiments by García-Martín et al. [13] revealed that ozone (1.6 mg kg^−1^) inhibited the mycelial growth of *Penicillium italicum* and *Penicillium digitatum*. Recently, Li et al. [14] reported that ozone (2 mg L^−1^) inhibited the diacetoxyscirpenol accumulation and disease incidence of *Fusarium sulphureum* in ‘Longshu No. 3′ potatoes during storage at 25 °C for seven days.

Ozone has been shown to be effective in controlling stem-end rot in horticultural crops. Terao et al. [2] reported that aqueous ozone (3 mg L^−1^) effectively controlled stem-end rot in ‘THB’ papaya during storage at 10 °C for seven days. Similarly, Minas et al. [15] revealed that ozone (0.3 µL L^−1^ for twenty-four hours) treatment reduced the disease incidence and severity of stem-end rot in kiwifruit stored at 0 °C for four months. The literature above shows that ozone has the potential to control stem-end rot in horticultural crops. However, there is limited literature reporting on the antifungal activity of O_3_ against stem-end rot (*Lasiodiplodia theobromae*) in mango fruit. Compared to the currently used treatments, ozone could provide an eco-friendly and economical technology to preserve fruit quality and control postharvest diseases. This research evaluated the effect of various ozone exposure times on mango fruit quality, enzyme activity, and *L. theobromae* incidence during storage.

## 2. Materials and Methods

### 2.1. Plant Material

Mango fruit (*cv*. Keitt) were harvested at physiological maturity from a commercial orchard at Goedgelegen Farm (25°778′ S, 30°447′ E) of Westfalia (Pty) LTD (Tzaneen, South Africa). Fruit were couriered overnight to the laboratory at the University of KwaZulu-Natal (Pietermaritzburg Campus, 29°624′ S, 30°403′ E). The fruit used in this experiment were disease-free, of the same color, size, and maturity. Each treatment had 54 fruit, replicated three times with three fruit per replicate.

### 2.2. Invitro Assays

#### 2.2.1. Isolation of *Lasiodiplodia theobromae*

The pathogen *L. theobromae* was isolated from infected mango fruit. Mango tissues were aseptically removed using a sterile scalpel. The infected tissue discs were sterilized for 30 s with ethanol (70%); after this, they were rinsed three times with distilled water. The discs were inoculated on potato dextrose agar (PDA) plates and incubated for seven days at 28 °C. The PDA medium was prepared by adding distilled water to 39 g of PDA, which was then autoclaved for 15 min at 121 °C, cooled down, and then poured into 90 mm Petri dishes.

#### 2.2.2. Mycelial Growth of *Lasiodiplodia theobromae*

The antifungal activity of gaseous ozone was determined as described by Tesfay et al. [16]. A mycelia disc (3 mm) was cut from *L. theobromae* growing colony (five days old) cultures and inoculated into the middle of a PDA plate. The Petri dishes were kept closed during O_3_ (0.25 mg/L) treatment, and the experiment was replicated three times, using three plates per replicate. The plates were stored at 90% relative humidity and 10 °C for fourteen days and thereafter transferred to 25 °C for seven days. Radial mycelial growth was measured with a ruler. The mycelial growth inhibition (MGI) percentage was calculated using Equation (1):MGI (%) = *D_c_* − *D_t_*/*D_c_* × 100(1)
where *D_t_* and *D_c_* are the mycelial growth diameter of the treated and control and Petri dishes, respectively.

### 2.3. In Vivo Assays

#### Fruit Artificial Inoculation

The spore suspension was prepared as described by Ong and Ali [12], with slight modifications. Briefly, two-week-old plates of *L. theobromae* were rinsed with 6 mL of distilled water. A sterile rod was used to remove the conidia on the plate surface. After this, the conidial count was adjusted to 10⁵ conidia mL^−1^ using a hemocytometer. Mango fruit were soaked in the conidial suspension for five minutes and dried at ambient air temperature for two hours. The study was conducted as a complete randomized design with three treatments, namely the untreated control, and ozone applied for 24 h (O_3_, 24 h) and 36 h (O_3_ 36 h).

### 2.4. Ozone Treatment

Mango fruit were treated with 0.25 mg/L gaseous ozone equipped with an ozone analyzer (Corona discharge ozone generator, Ozone Purification Technology, Johannesburg, South Africa) for twenty-four or thirty-six hours. Gaseous ozone was applied intermittently during cold storage at day zero and seven for twenty-four hours, and at day zero, seven, and fourteen for thirty-six hours. After two treatments (day 0 and 7), the fruit were removed from the ozone cold room and stored at 10 °C without ozone. The O_3_ times used in the current study were selected based on the biochemical parameters of the screening experiment done in 2018. The fruit were stored at 90% relative humidity and 10 °C for three weeks, mimicking shipment to European union export markets. After cold storage, the fruit were ripened at ambient temperature (25 °C) for seven days.

### 2.5. Fruit Sample Preparation

Mango samples were freeze-dried using a Vir Tis BenchTop Pro freeze drier (SP Scientific, Warminster, PA, USA). After this, the fruit samples were ground into powder and stored at −20 °C until use.

### 2.6. Fruit Mass Loss (%) and Firmness

Fruit firmness, expressed in Newtons (N), was calculated on the opposite side of the fruit cheeks using a handheld firmness tester (Model tip 0.25 cm^2^, Bareiss, Oberdischingen, Germany). Fruit were weighed on arrival at the laboratory and after every seven days. The same fruit were weighed throughout the storage period. The mass loss percentage was calculated using Equation (2):Weight loss (%) = *W*_0_ − *W*_1_/*W*_0_ × 100(2)
where *W*_0_ and *W*_1_ are the mass at day zero and at the end of storage, respectively.

### 2.7. Disease Index of Mango Fruit

Mango fruit were evaluated in terms of the disease index on the peel in each treatment. Fruit were classified according to their disease severity. Score 0: no visible decay, score 1: minor (1%) scattered decay, score 2: 2–20% decay incidence, score 3: 21–50% decay incidence, and score 4: more than 50% decay incidence. The DI index was calculated using Equation (3):DI (%) = Ʃ (DS × NFEC)/(TNF/HDS)(3)
where DS is the disease scale, NFEC is the number of fruit in each class, TNF is the total number of fruit, and HDS is the highest disease scale.

### 2.8. Color Analysis

Mango pulp color was determined using a chromameter (Chroma Meter, Konica Minolta Sensing, INC., Osaka, Japan) and expressed as chromaticity (c), Luminosity (L), a* values. The readings were done in triplicate on a half-cut mango surface. The chromameter was calibrated by scanning on a shite brick, y = 0.3215, Y = 87.0, X = 0.3.

### 2.9. Determination of Peroxidase Activity

Peroxidase (POD) enzyme extraction was performed as described by Criado et al. [17], with modification. In triplicate, sample powder (1 g) was extracted with 10 mL sodium phosphate buffer (0.2 M, pH 7.0). The extract was centrifuged (Avanti J-265 XP, Beckaman Coulter, Indianapolis, IN, USA) at 15,000 rpm for fifteen minutes at 4 °C. Enzyme extract (0.1 mL) was mixed with 2.7 mL sodium phosphate buffer (0.05 M), 1% p-phenylenediamine (0.2 mL), and 0.1 mL hydrogen peroxide (1.5% *w*/*v*). The absorbance was measured at 485 nm using a spectrophotometer (Shimadzu Scientific Instruments INC., Columbia, IN, USA). Enzyme activity was defined as an increase in absorbance at 485 nm per mg protein per minute.

### 2.10. Determination of Total Flavonoid Content

Flavonoids are secondary metabolites that play a vital role in defense against fungal pathogen attacks [18]. Total flavonoid content was measured as described by Eghdam and Sadeghi [19]. In triplicate, 0.5 g sample powder was extracted with 80% methanol at 40 °C for sixty minutes. Samples were cooled down and centrifuged (Avanti J-265 XP, Beckaman Coulter, Indianapolis, IN, USA) for fifteen minutes at 10,000 rpm. Fruit extract (0.1 mL) was added to 0.3 mL deionized water, followed by 0.03 mL sodium nitrate 5%, and incubated at 25 °C for five minutes. Aluminum chloride 10% (0.03 mL) was added; after five minutes, 0.2 mL sodium hydroxide (1 mM) was added to the solution, and the reaction mixture was diluted to 1 mL using deionized water. The absorbance was measured at 510 nm using a spectrophotometer (Shimadzu Scientific Instruments INC., Columbia, IN, USA) against methanol as a blank. Results were expressed as mg gallic acid equivalent (GAE)/g plant dry matter.

### 2.11. Statistical Analysis

The in vivo experiment was conducted using a randomized complete design with three treatments. Each treatment was replicated three times, using three fruit per replicate. Statistical analysis was performed using the analysis of variance (ANOVA) and means were separated at the 5% level using Fischer’s least significant difference (LSD). The GenStat statistical software (GenStat^®^, 18 edition, VSN International, Hemel Hempstead, UK) was used for the analysis.

## 3. Results

### 3.1. In Vitro Antifungal Assay of Gaseous Ozone against Lasiodiplodia theobromae

Gaseous ozone significantly (*p* < 0.05) inhibited the mycelial growth of *L. theobromae* compared to the control (Figure 1). There was no significant difference in the inhibition percentage of the O_3_ (24 h) and O_3_ (36 h) treatments.

### 3.2. Fruit Firmness

Fruit firmness significantly (*p* < 0.05) decreased with storage time in all three treatments (Table 1). Fruit treated with O_3_ (24 h) were firmer than in other treatments during storage. However, the firmness of ozone-treated fruit was not significantly different from day seven till the end of storage.

### 3.3. Fruit Mass Loss

Fruit mass loss increased significantly (*p* < 0.05) from day zero till the end of storage (Figure 2). The mass loss percentage of untreated fruit increased rapidly from day fourteen till the end of the storage time.

On day twenty-one, the treatment means of the control and O_3_ (24 h) were not significantly different. At the end of storage, fruit treated with O_3_ 36 h (19.63%) had lower mass loss than those subjected to O_3_ 24 h (25.39%) and the control treatment (29.20%).

### 3.4. Disease Index of Lasiodiplodia theobromae

The fruit disease incidence significantly (*p* < 0.05) increased during the storage time in both treated and untreated fruit (Table 2). The treatment means of O_3_ (24 h) and O_3_ (36 h) were not significantly different on days seven and fourteen. On day twenty-one, untreated fruit (68.33%) had a high disease incidence compared to O_3_ (24 h) (53.29%) and O_3_ (36 h) (43.30%). At the end of the storage time, fruit treated with O_3_ (36 h) had a low disease incidence compared to O_3_ (24 h) and the control.

### 3.5. Fruit Color

During the storage time, the luminosity decreased significantly (*p* < 0.05) in both treated and untreated fruit (Table 3). A sharp decrease in luminosity was observed in untreated fruit from day fourteen until the end of the storage time. Ozone-treated fruit showed a moderate reduction in luminosity at day twenty-one. The luminosity drastically declined in all treatments after cold storage.

At the beginning of storage, the fruit had a minimal greenish color. Color changes from greenish to orange-yellow were observed in all treatments during the storage time (Table 4). However, after fourteen days of storage, the untreated fruit showed rapid color changes. After twenty-one days of storage, no significant difference was observed between the treatment means of ozone-treated fruit. At the end of the storage time in the control (15.87), the fruit had a high a* value compared to O_3_ (24 h) and O_3_ (36 h), which had values of 10.04 and 8.91, respectively.

The chroma significantly (*p* < 0.05) varied between treatments. Chroma showed an increase in all treatments for the first two weeks of cold storage (Table 5). There was no significant difference between the treatment means of control and O_3_ (24 h) treatments at days fourteen and twenty-one. After cold storage, the chroma declined in all fruit. However, fruit treated with O_3_ (24 h) had high chroma compared to O_3_ (36 h) and the control.

### 3.6. Peroxidase Activity

The POD activity increased from day zero until day fourteen, and then decreased until the end of the storage period in all treatments (Figure 3). The POD activity of O_3_ (24 h) was significantly (*p* < 0.05) higher than that of other treatments from day fourteen till the end of the storage period.

The treatment means of O_3_ (36 h) and the control were not significantly different on day fourteen. At the end of storage, the POD activity of O_3_ 24 h (0.91 U min^−1^ g^−1^ DM) was high compared to O_3_ 36 h (0.80 U min^−1^ g^−1^ DM) and the control (0.78 U min^−1^ g^−1^ DM). The treatment means of O_3_ (24 h) and O_3_ (36 h) were not different at the end of the storage time.

### 3.7. Total Flavonoid Content

The total flavonoid content followed a similar trend to POD activity (Figure 4). On day fourteen, the highest flavonoid content was observed in O_3_ 36 h (144.49 mg GAE/g DM). The flavonoid content of fruit treated with O_3_ (36 h) was higher than in those subjected to other treatments from day fourteen until the end of the storage time. Untreated fruit had low flavonoid content compared to O_3_ (24 h) at the end of the storage time.

## 4. Discussion

The in vitro results revealed that *L. theobromae* is sensitive to gaseous ozone. Luo et al. [20] reported that ozone (79.44 ppm) inhibited the mycelial growth of *Penicillium expansum* and *Botrytis cinerea*. Similarly, ozone treatment (2 mg L^−1^) inhibited the colony diameter expansion of *Fusarium sulphureum* [21]. In the current study, the decreased mycelial growth could be attributed to O_3_ damaging the cell membrane. Liu et al. [21] reported that O_3_ increased the relative conductivity and lipid peroxidase, which destroyed the pathogen cell membrane. Ozone changes the morphology and ultrastructure, such as the cytoplasm and mitochondria of the pathogen [14]. Ozone reduces ergosterol accumulation and down-regulates the gene expression of the ergosterol biosynthesis pathway [14].

The current study showed that O_3_-treated fruit were firmer than untreated fruit. Similarly, Luo et al. [20] reported high firmness in ‘Hayward’ kiwifruit treated with ozone. Bambalele et al. [22] reported that ozone (0.25 mg/L) retained firmness in ‘Keitt’ mango during storage. Ozone has the potential to be combined with edible coatings or hot water and used as a postharvest treatment for horticultural crops. Ozone combined with hot water retained firmness in ‘Carbarosa’ strawberries during storage at 1 °C [23]. The retained firmness could be attributed to O_3_ inhibiting cell-wall-degrading enzymes. Ozone inhibits the enzyme activity of *β*-D-galactopyranosidase, Glycosidase *α*-L-arabinopyranosidase, and polygalacturonase in ‘Caldeo’ cantalope melon during storage [24]. The retained fruit firmness could be attributed to ozone decreasing ethylene production. These findings indicate that ozone maintained firmness and delayed fruit ripening.

Mass determines the price of fresh produce, hence being a vital parameter. Ozone had no effect on mass loss in ‘Hicaznar’ pomegranate during storage [25]. Bambalele et al. [26] observed low mass loss in mango fruit treated with ozone. Similarly, ozone reduced mass loss in ‘Daw’ longan fruit during storage [27]. The treatment combination of moringa + CMC + ozone reduced mass loss in ‘Hass’ avocado during storage at 5.5 °C [28]. The reduced mass loss could be attributed to the ozone altering permeability and transpiration in the fruit.

The untreated fruit showed more rapid disease incidence than ozone-treated fruit. Chamnan et al. [27] observed low disease incidence in longan fruit treated with a high level of ozone. Gaseous ozone decreased the disease incidence of *Penicillium expansum* and *Botrytis cinerea* in kiwifruit during cold storage [20]. Ozone and hot water treatments controlled stem-end rot in ‘THB’ papaya during storage [2].

Color is related to fruit quality because consumers purchase fresh produce based on appearance. In the present study, gaseous ozone treatment had a significant effect on mango color. Mango color changes from green to orange-yellow were delayed until day fourteen. Shezi et al. [28] observed similar results in ‘Hass’ avocados treated with ozone during storage at 5.5 °C for twenty-eight days. There was a decrease in L* and chroma values, whereas the a* values decreased at the end of storage. These results indicate the augmented intensity of orange-yellow mango color and fruit ripening. Similarly, Panou et al. [23] reported an increase in a* and a decline in the L* and b* values of ‘Carborosa’ strawberries during storage at 1 °C for fifteen days. The increased a* could be attributed to chlorophyll degradation and carotenoid biosynthesis during mango fruit ripening.

Ozone treatment enhanced the POD activity in kiwifruit during storage [20]. Terao et al. [2] reported that aqueous ozone did not affect the POD activity in papaya inoculated with *L. theobromae.* POD is a defense enzyme that plays a vital role in identifying the plant’s disease resistance [20,29]. POD controls relative oxygen species (ROS) levels during stress conditions, thus protecting the cell against pathogen invasion [12,30]. In the current study, the enhanced disease resistance of stem-end rot could be attributed to ozone up-regulating ROS-related genes. Ozone up-regulated 2-Cys Prx genes in apples inoculated with *Botrytis cinerea* during storage [30].

The results of total flavonoid content are similar to those previously reported by Zhu et al. [31]. The authors observed an increase of 11.30% in the flavonoid content of ‘Satsuma’ mandarins treated with ozone. Mustapha et al. [32] reported that O_3_ enhanced the flavonoid content in ‘Jutou’ cherry tomatoes during storage. The enhanced flavonoid content could be attributed to the enzyme activity of phenylalanine ammonium lyase (PAL). Gaseous ozone increased the gene expression of PAL, chalcone isomerase, and chalcone synthase in satsuma mandarin during storage [31]. Chalcone isomerase and chalcone synthase are the key enzymes involved in the flavonoid biosynthesis pathway [33]. Ozone could induce oxidative stress in plants. The enhanced flavonoid content could be attributed to induced enzyme activity during biotic conditions.

## 5. Conclusions

This study revealed that gaseous ozone effectively controlled the stem-end rot in mango fruit. Ozone inhibited the mycelial growth of *L. theobromae*, maintained fruit firmness, and delayed mass loss during storage. Gaseous ozone (24 h) maintained the total flavonoid content and POD activity in mango fruit during storage. There was no significant difference between the treatment means of O_3_ (24 h) and O_3_ (36 h) in terms of firmness, -color, and POD activity during storage. This suggests that increasing the O_3_ exposure time did not affect these parameters. Therefore, O_3_ (24 h) is recommended as an effective postharvest treatment to control stem-end rot and enhance the quality of mango fruit.

## Figures and Tables

**Figure 1 foods-12-00195-f001:**
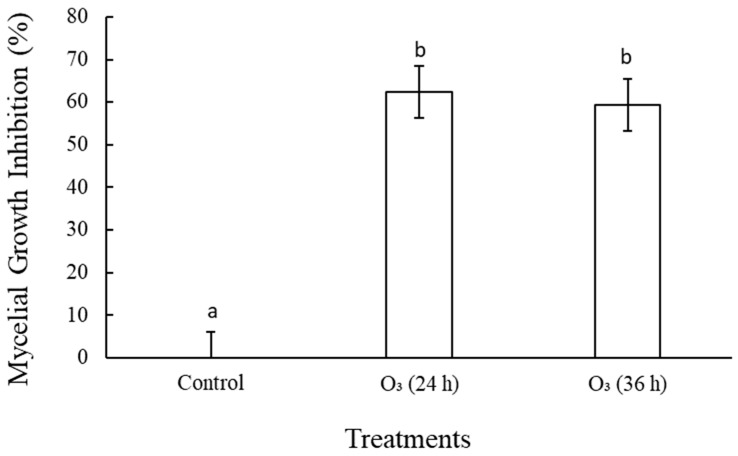
Effect of gaseous ozone (0.25 mg/L) on mycelial growth inhibition (%) of *L. theobromae* incubated at 10 °C for fourteen days and seven days at 25 °C (*p* < 0.05; least significance difference = 12.27; *n* = 9). Bars with the same letter are not significantly different (*p* < 0.05).

**Figure 2 foods-12-00195-f002:**
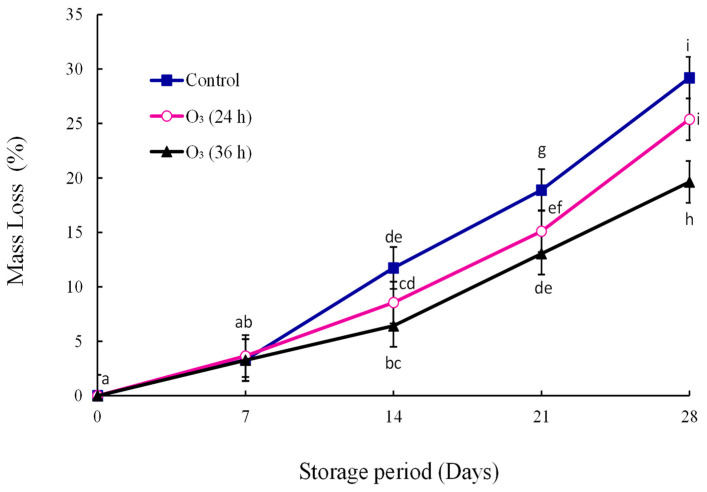
Effect of gaseous ozone (0.25 mg/L) on mango fruit mass loss (%) inoculated with *L. theobromae*, stored at 10 °C for three weeks and ripened at ambient temperature for seven days (*p* < 0.05; ±SE; *n* = 9). The letters a–i are used above or below the reported values. The means with the same letter are not significantly different (least significant difference = 1.92).

**Figure 3 foods-12-00195-f003:**
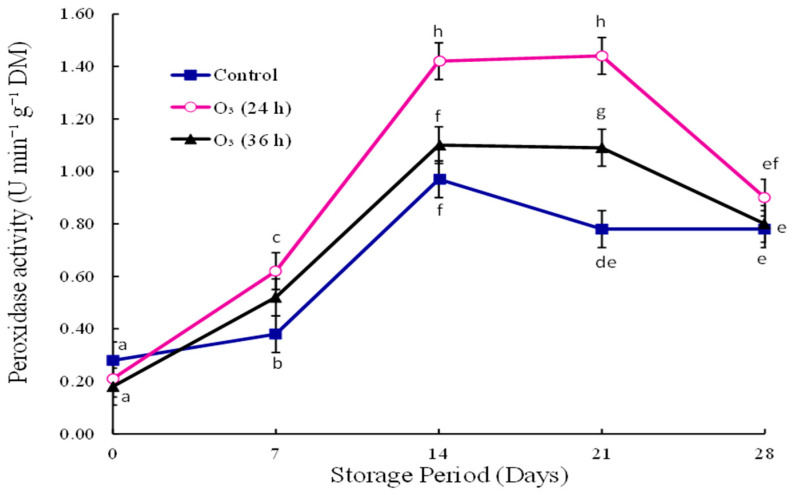
Effect of gaseous ozone (0.25 mg/L) on peroxidase activity in *L. theobromae*-inoculated mango fruit, stored at 10 °C for three weeks and ripened at ambient temperature for seven days (*p* < 0.05; ±SE; *n* = 9). The letters a–h are used above or below the reported values. The means with the same letter are not significantly different (least significant difference = 0.14).

**Figure 4 foods-12-00195-f004:**
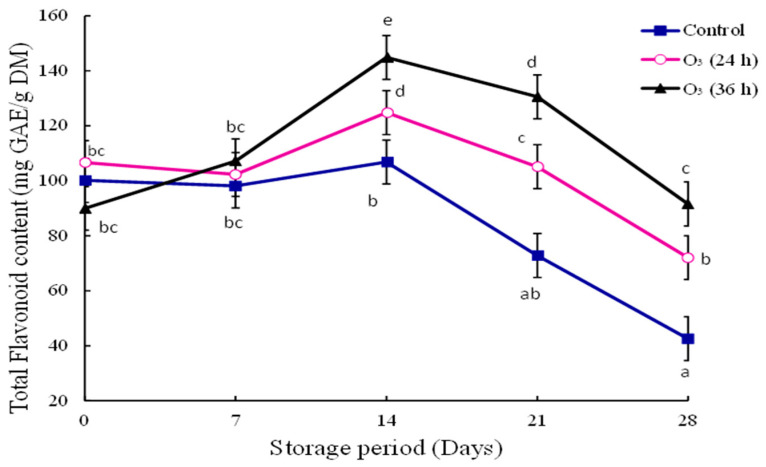
Effect of gaseous ozone (0.25 mg/L) on total flavonoid content of mango fruit inoculated with *L. theobromae*, stored at 10 °C for three weeks and ripened at ambient temperature for seven days (*p* < 0.05; ±SE; *n* = 9). The letters a–e are used above or below the reported values. The means with the same letter are not significantly different (least significant difference = 15.95).

**Table 1 foods-12-00195-t001:** Effect of gaseous ozone (0.25 mg/L) on the firmness of mango fruit inoculated with *L. theobromae*, stored at 10 °C for three weeks and ripened at ambient temperature for seven days.

Treatment	Storage Period (Days)				
	0	7	14	21	28
Control	48.36 ^hi^	42.69 ^f^	38.85 ^cd^	32.59 ^b^	25.90 ^a^
O_3_ (24 h)	51.20 ^hi^	46.86 ^gh^	42.86 ^f^	38.85 ^cde^	32.34 ^b^
O_3_ (36 h)	48.78 ^i^	45.02 ^fg^	41.98 ^df^	37.94 ^c^	32.72 ^b^

Treatment means with the same letter between rows and within columns are not significantly different (*p* < 0.001; least significance difference = 1.36; cv = 8.00; *n* = 9).

**Table 2 foods-12-00195-t002:** Effect of gaseous ozone (0.25 mg/L) on disease incidence of mango fruit inoculated with *L. theobromae*, stored at 10 °C for three weeks and ripened at ambient temperature for seven days.

Treatment	Storage Period (Days)				
	0	7	14	21	28
Control	0.00 ^a^	10.00 ^b^	24.12 ^c^	68.33 ^f^	94.44 ^h^
O_3_ (24 h)	0.00 ^a^	8.53 ^ab^	17.02 ^bc^	53.29 ^e^	81.11 ^g^
O_3_ (36 h)	0.00 ^a^	8.73 ^ab^	15.71 ^bc^	43.30 ^d^	70.67 ^f^

The means with the same letter between rows and within columns are not significantly different (*p* < 0.05; least significance difference = 8.80; cv = 18.50; *n* = 9).

**Table 3 foods-12-00195-t003:** Effect of gaseous ozone (0.25 mg/L) on L* values of on mango fruit inoculated with *L. theobromae*, stored at 10 °C for three weeks and ripened at ambient temperature for one week.

Treatment	Storage Period (Days)				
	0	7	14	21	28
Control	70.52 ^i^	67.92 ^hi^	55.39 ^e^	39.31 ^b^	27.87 ^a^
O_3_ 24 h	69.33 ^i^	65.13 ^gh^	60.41 ^f^	49.60 ^d^	38.24 ^b^
O_3_ 36 h	69.43 ^i^	67.37 ^hi^	62.04 ^fg^	56.67 ^e^	45.25 ^c^

The means with the same letter between rows and within columns are not significantly different (*p* < 0.05; least significant difference = 3.30; *n* = 9; cv% = 6.20).

**Table 4 foods-12-00195-t004:** Effect of gaseous ozone (0.25 mg/L) on a* values of mango fruit inoculated with *L. theobromae*, stored at 10 °C for three weeks, and ripened at ambient temperature for one week.

Treatment	Storage Period (Days)				
	0	7	14	21	28
Control	−1.45 ^a^	3.14 ^bc^	7.29 ^ef^	9.85 ^g^	15.87 ^h^
O_3_ (24 h)	−1.59 ^a^	2.79 ^b^	4.72 ^cd^	7.55 ^ef^	10.04 ^g^
O_3_ (36 h)	−1.36 ^a^	3.98 ^bcd^	4.91 ^d^	6.85 ^e^	8.91 ^fg^

The means with the same letter between rows and within columns are not significantly different (*p* < 0.001; least significant difference = 1.56; *n* = 9; cv% = 13.08).

**Table 5 foods-12-00195-t005:** Effect of gaseous ozone (0.25 mg/L) on chroma of mango fruit inoculated with *L. theobromae*, stored at 10 °C for three weeks and ripened at ambient temperature for one week.

Treatment	Storage Period (Days)				
	0	7	14	21	28
Control	57.79 ^a^	65.12 ^cd^	61.39 ^abc^	68.82 ^de^	59.86 ^ab^
O_3_ (24 h)	63.48 ^bc^	69.02 ^de^	65.33 ^cd^	69.63 ^d^	65.26 ^cd^
O_3_ (36 h)	58.87 ^a^	65.33 ^cd^	60.22 ^ab^	72.48 ^e^	64.64 ^c^

The means with the same letter between rows and within columns are not significantly different (*p* < 0.05; least significant difference = 3.49; *n* = 9; cv% = 5.80).

## Data Availability

Data is included in this article.

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
