# Peer review of "The Antifungal Effect of Gaseous Ozone on Lasiodiplodia theobromae Causing Stem-End Rot in ‘Keitt’ Mangoes"

_foods, 2023, doi:10.3390/foods12010195_

Round 1

Reviewer 1 Report

This article aimed to investigate the antifunal effect of gaseous ozone on Lasiodiplodia theobromae causing stem-end rot in ‘Keitt’ mangoes, but there were several problems needed to explain.

>2.2.2 How did mycelia disc of L. theobromae was treated with O3? The concentration of O3 was not mentioned in this part.

>2.4 In the Mango storage experiment, Mango was stored in low temperature for 21 days and then was transferred to ambient temperature for 7 days. But in the antifungal activity of gaseous ozone experiment, mycelia disc of L. theobromae was cultured at 10 ℃ for fourteen days and thereafter transferred to 25 ℃ for seven days. Why are the two times different?

>In addition, why the ozone treatment was applied intermittently at day 0,7 for 24 h and at day 0, 7,14 for 36 h? Ozone treated Mango at day 14 for 24 h was missed. So the results about the two experimental groups couldn’t be compared scientifically.

>How did the author ensure the amount of ozone used in both treatment groups is the same? 0.25 mg/L gaseous ozone for 24 h or 36 h? After that, gaseous ozone was disappeared? Please accurately describe the amount of ozone of different treatment groups.

>The appearance, color value and respiratory rate of all Mango groups need to be supplemented.

>L117 the expression of storage temperature has problem.

>Can storage for three weeks mimick shipment to European union export markets? Maybe the author omitted one important factor-vibration during the transportation.

>L187 reference was missed.

>Figure 2 add the significance among different groups. Why Mango fruit treated with O3 36 h has lower mass loss than other groups?

>Please delete the 2.3.1.

>Please check the mistakes in the references.

Author Response

Dear Editor

Thank you for sending us feedback and comments from the reviewers. We have addressed all the comments and made the necessary changes. The new alterations are highlighted in red color.  

The following is a detailed response to reviewers’ comments:

Reviewer #1:

Comment: 2.2.2 How did mycelia disc of L. theobromae was treated with O3? The concentration of Owas not mentioned in this part.

Response: We thank the reviewer for the comment. As suggested by the reviewer, the ozone concentration was added; please refer to line 96.

Comment: 2.4 In the Mango storage experiment, Mango was stored in low temperature for 21 days and then was transferred to ambient temperature for 7 days. But in the antifungal activity of gaseous ozone experiment, mycelia disc of L. theobromae was cultured at 10 â„ƒ for fourteen days and thereafter transferred to 25 â„ƒ for seven days. Why are the two times different?

Response: We thank the reviewer for the comment. The ambient temperature used for the study was 25 ℃. The temperature time was corrected; please refer to line 126. The in vitro  strorage time used for the current experiment  is supported by Barbosa-Martínez et al., 2002, Terao et al., 2019, Liu et al., 2020 & Shah et al., 2021. In their studies the invitro storage time ranges from 5 to 21 days and it is different to the in vivo storage period.

Comment: In addition, why the ozone treatment was applied intermittently at day 0,7 for 24 h and at day 0, 7,14 for 36 h? Ozone treated Mango at day 14 for 24 h was missed. So the results about the two experimental groups couldn’t be compared scientifically.

Response: We thank the reviewer for the comment. The objective of the current study was to evaluate the effect of various ozone exposure times on stem-end rot in mango fruit. The treatments were the control, ozone exposure for 24  or 36 hours. Applying ozone at day fourteen would have increased ozone time from 24 to 36. This would make both ozone treatments similar.  

Comment: How did the author ensure the amount of ozone used in both treatment groups is the same? 0.25 mg/L gaseous ozone for 24 h or 36 h? After that, gaseous ozone was disappeared? Please accurately describe the amount of ozone of different treatment groups.

Response: We thank the reviewer for the comment. The ozone generator used is equipped with the analyzer which was used to monitor the ozone concentration. The fruits were removed from the ozone cold room after 24-hour treatment and stored in different cold storage; please refer to lines 118-119; 122-123.

Comment: The appearance, color value and respiratory rate of all Mango groups need to be supplemented.

Response: We thank the reviewer for the comment. This experiment is part of an ongoing study, and the respiration rate results were not included as they are part of another study. The color parameters were added; please refer to lines  154-158; 247-291; 352-361.

Comment: L117 the expression of storage temperature has problem.

Response: We thank the reviewer for the comment. The temperature was corrected.

Comment: Can storage for three weeks mimick shipment to European union export markets? Response: We thank the reviewer for the comment. The shipment of fresh produce from South Africa to European markets takes three weeks. The cold storage time used for this study was described by Sivakumar et al., 2011, Shezi et al., 2020, and Kubheka et al., 2020.

Comment: Maybe the author omitted one important factor-vibration during the transportation.

Response: We thank the reviewer for the comment. The fruit were packed in cardbox boxes transportation. The fruits were handled with care to ensure that no injuries and bruises occured.

Comment: L187 reference was missed.

Response: We thank the reviewer for the comment. Line 187 describes the fruit firmness results; hence no reference was included.

Comment: Figure 2 add the significance among different groups. Why Mango fruit treated with O3 36 h has lower mass loss than other groups?

Response: We thank the reviewer for the comment. The significant difference between treatment means is represented by error bars in the graph. The possible mode of action of ozone on fruit mass loss is explained in the discussion section; please refer to lines  340-346.

Comment: Please delete the 2.3.1.

Response: We thank the reviewer for the comment. As suggested by the reviewer, this was deleted.

Comment: Please check the mistakes in the references.

Response: We thank the reviewer for the comment. All mistakes were corrected, please refer to lines 415-488. 

Reviewer 2 Report

1.      Line 12 mango fruits? please check the grammar throughout the MS. Similar mistakes in line 13

2.      In introduction, please write the background and origin of problem

3.      Line 21 please write recommendations

4.      In introduction please write earlier work published on the same topic and how your work is novel in this regards.

5.      Write clear objectives

6.      Line 68 fruits samples, choose of the word is also very important for this MS

7.      Line 77 Lasioderma full

8.      Line 89 experiments were repeated four times, what about repetition of experiments

9.      Please provide symptomatic image of each disease index

10.  Data Analysis: The major drawback of this paper is data analysis, author should add a statistical section, whether the experiment was repeated, if repeated then how did they handle data from repeated experiment, if they used combine data, then did they perform homogeneity test to check the significant level ? If you did separate analysis, justify the reason for that etc. further is not clear either they used raw data or transform data for analysis? in this sense, did they check normality test ? for normality test (whether data transformation was required based on normality test, if so what data transformation techniques {such as log transformation/square root/arc-sign transformation) was used to transfer data. Also nothing about the ANOVA analysis, If the collected data were substantially different from ANOVA, the averages should be compared using Tukey's HSD test. These all things are missing in this MS. If the experiments were not repeated the MS can be rejected

11.  Although, authors discuss results, but the data in results presented should be supported with statistical value.

12.  Discussion is very poorly written, please discussion the mode of mechanism involve in management, and how your technology can be integrated with other methods to manage this major mango disease

13.  Provide the image of treated and control fruits showing symptoms of fruit rots

14.  In each table add CV value and mention full name of LSD

Author Response

Dear Editor

Thank you for sending us feedback and comments from the reviewers. We have addressed all the comments and made the necessary changes. The new alterations are highlighted in red color.  

The following is a detailed response to reviewers’ comments:

Reviewer #2:

Comment: Line 12 mango fruits? please check the grammar throughout the MS. Similar mistakes in line 13

Response: We thank the reviewer for the comment. For this manuscript, the fruit was changed to fruits, Please refer to lines 12 &13.

Comment: In introduction, please write the background and origin of problem

Response: We thank the reviewer for the comment. After carefully considering the comment by our reviewer, we believe that the background and problem statement for this study is provided in Line  28 to 39 of the manuscript.

Comment: Line 21 please write recommendations

Response: We thank the reviewer for the comment. The recommendations were added; please refer to lines 23.

Comment: In introduction please write earlier work published on the same topic and how your work is novel in this regards.

Response: We thank the reviewer for the comment. Literature on as postharvest technology of fresh produce was added; please refer to lines 62-73.

Comment: Write clear objectives

Response: We thank the reviewer for the comment. The objective was rewritten 71-73.

Comment: Line 68 fruits samples, choose of the word is also very important for this MS

Response: We thank the reviewer for the comment. The wording was changed, please refer to line 76.

Comment: Line 77 Lasioderma full

Response: We thank the reviewer for the comment. As suggested by the reviewer, L. theobromae was written in full, please refer to lines 85; 94;  195; 236.

Comment: Line 89 experiments were repeated four times, what about repetition of experiments

Response: We thank the reviewer for the comment. The mistake was corrected. The experiment was replicated three times using three plates per replicate; please refer to lines 98.

Comment: Please provide symptomatic image of each disease index

Response: We thank the reviewer for the comment. The images of artificially inoculated mango fruit was added, please refer to lines 206-211 .

Comment: Data Analysis: The major drawback of this paper is data analysis, author should add a statistical section, whether the experiment was repeated, if repeated then how did they handle data from repeated experiment, if they used combine data, then did they perform homogeneity test to check the significant level ? If you did separate analysis, justify the reason for that etc. further is not clear either they used raw data or transform data for analysis? in this sense, did they check normality test ? for normality test (whether data transformation was required based on normality test, if so what data transformation techniques {such as log transformation/square root/arc-sign transformation) was used to transfer data. Also nothing about the ANOVA analysis, If the collected data were substantially different from ANOVA, the averages should be compared using Tukey's HSD test. These all things are missing in this MS. If the experiments were not repeated the MS can be rejected

Response: We thank the reviewer for the comment. Statistical analysis was used for the current manuscript. Details of the treatment structure and type of statistical design used have been furnished in Line 188 to 193 of the revised manuscript. Further details were provided under different sections of the manuscript (for example, Lines 81-83; 98; 115-117; 164, 176). Additionally, for each picture cation and table footer, the sample size was described; please refer to lines 205, 222; 230-231; 247, 304, 320.

Comment: Although, authors discuss results, but the data in results presented should be supported with statistical value.

Response: We thank the reviewer for the comment. We have reviewed the manuscript and believe that statistical data was provided in the results section . The statistical data provided includes the p-value, LSD and sample size, please refer to lines 194-320.

Comment: Discussion is very poorly written, please discussion the mode of mechanism involve in management, and how your technology can be integrated with other methods to manage this major mango disease

Response: We thank the reviewer for the comment. The possible mode of action of the current treatment wa added, please refer to lines 334-336; 339-340; 345-346.

Comment: Provide the image of treated and control fruits showing symptoms of fruit rots

Response: We thank the reviewer for the comment. The images were added; please refer to lines 206-212.

Comment: In each table add CV value and mention full name of LSD

Response: We thank the reviewer for the comment. The cv value was added and LSD was written in full; please refer to lines 222 & 247. 

Round 2

Reviewer 1 Report

The author has revised all problems and answered all comments. I have no questions.

Author Response

We thank the reviewer for the helpful comments as they improved the quality of our manuscript. This is well-appreciated.